



# Changes of Anthropogenic Precursor Emissions Drive Shifts of Ozone Seasonal Cycle throughout Northern Midlatitude Troposphere

Henry Bowman[1], Steven Turnock[2,3], Susanne E. Bauer[4,5], Kostas Tsigaridis[5,4], Makoto Deushi[6], Naga Oshima[6], Fiona M. O'Connor[2], Larry Horowitz[7], Tongwen Wu[8], Jie Zhang[8], and David D. Parrish[9]

[1]Carleton College, Northfield, MN, 55057, USA
   [2]Met Office Hadley Centre, Exeter, UK
   [3]University of Leeds Met Office Strategic (LUMOS) Research Group, School of Earth and Environment, University of Leeds, UK
   [4]NASA Goddard Institute for Space Studies, New York, NY, USA
[5]Center for Climate Systems Research, Columbia University, New York, NY, USA
   [6]Meteorological Research Institute, 1-1 Nagamine, Tsukuba, Ibaraki, 305-0052, Japan
   [7]NOAA Geophysical Fluid Dynamics Laboratory, Princeton, NJ, 08450, USA
   [8]Beijing Climate Center, China Meteorological Administration, Beijing, China
   [9]David.D.Parrish LLC, Boulder, CO, 80303, USA

*Correspondence to*: Henry Bowman (bowmanh@carleton.edu), David D. Parrish (david.d.parrish.llc@gmail.com)

**Abstract.** Simulations by six CMIP6 Earth System Models indicate that the seasonal cycle of baseline tropospheric ozone at northern midlatitudes has been shifting since the mid-20[th] Century. Beginning in ~1940 the seasonal cycle increased in amplitude by ~10 ppb (measured from seasonal minimum to maximum), and the seasonal maximum shifted to later in the

year by about 3 weeks. This shift maximized in the mid-1980s, followed by a reversal - the seasonal cycle decreased in amplitude and the maximum shifted back to earlier in the year. Similar changes are seen in measurements collected from the 1970s to the present. The timing of the seasonal cycle changes is generally concurrent with the rise and fall of anthropogenic emissions that followed industrialization and subsequent implementation of air quality emission controls. We quantitatively compare the temporal changes of the ozone seasonal cycle at sites in both Europe and North America with the temporal

changes of ozone precursor emissions across the northern midlatitudes and find a high degree of similarity between these two temporal patterns. We hypothesize that changing precursor emissions are responsible for the shift in the ozone seasonal cycle, and suggest the mechanism by which changing emissions drive the changing seasonal cycle: increasing emissions of $NO_X$ allow summertime photochemical production of ozone to become more important than ozone transported from the stratosphere and increasing VOCs lead to progressively greater photochemical ozone production in the summer months,

thereby increasing the amplitude of the seasonal ozone cycle. Decreasing emissions of both precursor classes then reverse these changes. The quantitative parameter values that characterize the seasonal shifts provide useful benchmarks for evaluating model simulations, both against observations and between models.



## 1 Introduction

Tropospheric ozone is a harmful air pollutant and greenhouse gas. It is a secondary pollutant, formed as a photochemical
product of oxidation reactions involving volatile organic compounds (VOCs), carbon monoxide (CO), and methane ($CH_4$) in
the presence of oxides of nitrogen ($NO_X$). Entrainment of stratospheric ozone also contributes to tropospheric ozone
concentrations. Ozone is lost from the troposphere to surface deposition and additional photochemical reactions. The
processes driving ozone formation and destruction are complex, which adds difficulty to the task of understanding the
impacts of tropospheric ozone on human and ecosystem health and climate change. Because ozone is not directly emitted,
areas of ozone formation and enhanced concentrations are often geographically separated from emission sources. The
lifetime of ozone in the troposphere is long enough - approximately 22 days averaged globally (Young et al., 2013) - that it is
transported over hemispheric scales. Its lifetime is even longer above the planetary boundary layer (PBL) due to slower
losses in the free troposphere (FT) and continuing formation from transported precursors (Fowler et al., 2008). At northern
midlatitudes (defined here as between 30° and 60° N), prevailing westerly winds and the long lifetime of ozone result in a
high degree of zonal similarity in baseline ozone concentrations (Chan et al., 2010; Parrish et al., 2014), with similar
temporal changes in baseline ozone observed at multiple sites throughout that zone (Cooper et al., 2014; Parrish et al., 2020).
In this work we use the term "baseline" to denote air that has not been influenced by direct, recent continental influences -
see discussion in Chapter 1 of Hemispheric Transport of Air Pollution (HTAP, 2010). Another consequence of rapid
transport of ozone and its precursors is that emissions from any location in the northern midlatitude region can influence
ozone concentrations throughout the zone. Depending on emissions upwind of a particular site, that site may be
representative of only baseline ozone conditions, or a combination of baseline conditions and regional or local processes.

At northern midlatitudes, outside of the marine boundary layer (MBL), tropospheric ozone follows a seasonal cycle with
annual maximum concentrations in late spring or early summer, due to peak stratospheric influence in late winter or spring,
peak photochemical production in the summer (e.g., Logan et al., 1985), and a summertime emission maximum of the
important biogenic VOC precursors (e.g., Guenther et al., 1995). Within the MBL, ozone has a summertime minimum due to
the much faster photochemical ozone losses in that season and the absence of strong photochemical production due to
limited $NO_X$ emissions in that environment. However, the seasonal ozone cycle has not been constant over time; many
previous studies have noted a shift in the seasonal ozone cycle. These studies have been measurement- and model-based,
cover northern midlatitude locations in Europe and North America and describe shifts in either the amplitude or phase of the
seasonal ozone cycle. For example, a seasonal cycle shift at Hohenpeissenberg, Germany has been identified, based on
observations, in which ozone reached its annual maximum in the summer in the 1970s, but now ozone is nearly equal
between the spring and summer seasons (Parrish et al., 2012; 2013). Other studies found similar shifts in the timing of the
annual ozone maximum at other European sites (Parrish et al., 2012; Cooper et al., 2014 and references therein), the
northeastern and eastern US (Bloomer et al., 2010; Clifton et al., 2014), California (Cooper et al., 2014; Parrish et al., 2017),
and the western US (Cooper et al., 2012). Other papers have indirectly provided evidence of a shift in the phase of the



seasonal ozone cycle without expressly mentioning this shift. For example, studies have documented increasing springtime ozone (Lin et al., 2015), often in combination with decreasing summer ozone (Chan et al., 2010; Cooper et al., 2012; Lin et al., 2017), which indicates that the seasonal cycle is shifting towards a springtime maximum.

Some studies discuss another aspect of the varying seasonal ozone cycle: changes in its amplitude. This finding is most
evident in measured data collected across the US (Simon et al., 2014), and specifically in the eastern US (Strode et al., 2015). Other studies do not explicitly mention the changing amplitude but still provide evidence for this phenomenon. According to measurements and models, these papers find ozone decreasing in summer, when it has typically been highest (Hogrefe et al., 2011; Cooper et al., 2012; Parrish et al., 2012; 2013 and references therein; Simon et al., 2014; Lin et al., 2017), and concurrently increasing in the winter, when it has typically been lowest (Bloomer et al., 2010; Chan et al., 2010;
Cooper et al., 2012; Parrish et al., 2012; Lin et al., 2017). In combination, these changes imply that the amplitude of the seasonal ozone cycle has decreased from its level in the 1990s, a time period characterized by decreasing precursor emission concentrations across northern midlatitudes.

Ozone precursor emission changes have been hypothesized as the cause of shifts in the seasonal ozone cycle in some studies, which have reached a consensus about how emissions affect the seasonal cycle. In the absence of large
anthropogenic precursor emissions, the seasonal ozone maximum occurs in the spring (Logan et al., 1985; Cooper et al., 2014 and references therein). Higher emissions correspond to a seasonal cycle of larger magnitude with seasonal maximum ozone occurring later in the year (in summer); likewise, lower emissions correspond to a seasonal cycle of smaller amplitude with an earlier-occurring spring maximum (Parrish et al., 2013; Clifton et al., 2014; Cooper et al., 2014; Strode et al., 2015). Most of these studies investigate areas affected only by baseline conditions or areas where local emissions have been
controlled in recent years, and thus capture the decrease in amplitude and shift towards an earlier maximum. However, there is some analysis of areas with increasing emissions, and the seasonal cycle grew in amplitude with a progressively later maximum. For example, $NO_X$ emissions roughly tripled since 1990 in parts of China, and summertime ozone has increased at many polluted sites (e.g., sites directly downwind of these increasing emissions) by up to 2 ppb/year (Li et al., 2017); thus on local to regional scales in China, increasing emissions correspond to a growing seasonal cycle with a shift towards
summer. Most of the European and North American studies were largely based on observations and simulations from the late 20th Century and early 21st Century, when emissions were generally decreasing. One exemption is the study of Marenco et al. (1994) that noted the preindustrial 19th Century seasonal maximum occurred in the spring at a remote European site, but that maximum had shifted towards the summer by the 1980s. Taken together, these results suggest that the increase of anthropogenic precursor emissions during industrial development shifts the ozone seasonal cycle toward the summer, and
reductions in those emissions allow the seasonal cycle to shift back toward the preindustrial condition.

Other studies identify correlations between precursor emissions and a changing seasonal cycle at sites separated geographically (instead of at the same site studied across a period of time). For example, sites in eastern Canada are subject to less pollution than sites in the eastern US and subsequently show smaller summertime and larger wintertime ozone concentrations, evidence that the amplitude of the seasonal cycle is smaller in the absence of precursor emissions (Chan et



al., 2010). Across the same sites, a springtime ozone maximum is observed for more pristine Canadian sites, while the more polluted eastern US displays a summertime maximum (Chan et al., 2009).

A quantitative understanding of the link between precursor emissions and the seasonal ozone cycle will benefit air quality policy development. Reider et al. (2018) note that changes to the seasonal ozone cycle may influence the timing and number of days of ozone exceedance above the US National Ambient Air Quality Standard (NAAQS). As such, understanding the

seasonal cycle - including how it changes in response to changing emissions - may usefully inform air quality control managers across the world in setting future ozone standards in efforts to reduce the harmful impacts of surface ozone (Lin et al., 2017). A changing future climate will bring further uncertainty, including the possibility of an ozone–climate penalty (Rasmussen et al., 2013); we can abate some of this uncertainty by understanding the interactions between emissions and atmospheric impacts (e.g., the seasonal ozone cycle) as fully as possible.

Despite the extensively documented record of shifts in the seasonal ozone cycle, no previous study has quantitatively analyzed measured data and model simulation results from across the northern midlatitude region, examined shifts in the amplitude and phase of the seasonal ozone cycle, quantitatively analyzed changing precursor emissions alongside seasonal cycle shifts, and proposed the mechanisms by which changing emissions affect the seasonal cycle; this paper aims to accomplish these tasks. We examine sites representative of baseline conditions in both Western Europe and North America.

Given the zonal similarity of ozone at northern midlatitudes, our analysis is expected to be representative of the baseline troposphere throughout northern midlatitudes. We investigate seasonal ozone cycle changes that began ~75 years ago, before reliable ozone measurements are available; thus, we rely on historical simulations from Coupled Model Intercomparison Project Phase 6 (CMIP6) Earth System Models (ESMs) as our primary basis for seasonal ozone cycle analysis. We compare these simulation results to available observations. A previous study of seasonal ozone cycle found that the previous

generation of Earth system models poorly simulated the seasonal cycle, including changes to it (Parrish et al., 2013). However, Griffiths et al. (2020) find that CMIP6 ESMs capture the general shape of the observed seasonal ozone cycle averaged between 30° and 90° N, despite a positive bias of 3-4 ppb in overall ozone concentrations. Thus, CMIP6 ESMs may be more reliable for ozone seasonal cycle analysis than previous models.

In this work we investigate two quantities that define the seasonal cycle of tropospheric ozone: the amplitude (the

difference between the annual average and the annual maximum or minimum ozone concentrations) and the phase (the timing of annual maximum ozone concentrations). Model simulations indicate that both of these quantities have changed over past decades; we compare their shifts with temporal changes in ozone precursor emissions that are prescribed in the models. In the following sections, we describe our analytical methods, present the analysis results, and discuss those results within a broader context. The overall goal of the paper is to provide a quantitative analysis of the shifting seasonal cycle of

tropospheric ozone at northern midlatitudes. Since the seasonal cycle reflects the sources and loss of ozone, quantifying it provides the opportunity for comparison of the simulated seasonal cycle between different models, and for comparison of model simulations with the limited record of observations. Comparing models and observations is an important way to gain insight into the performance of the models.


## 2 Methods

In this work, we seek to quantify the ozone seasonal cycle based on a small set of parameter values that reflect the amplitude and phase of that cycle. To accomplish this quantification, we analyze monthly mean ozone concentrations from ESM simulations as well as observations, to the extent they are available. Monthly means have sufficient temporal resolution to capture seasonal changes, while effectively averaging over most variability driven by diurnal and meteorological changes. Our goal is to investigate the long-term changes in the seasonal cycle over the 1850-2014 period included in the CMIP6

historical simulations. Ozone varies systematically on decadal scales, and also has temporal variability on interannual scales (i.e., on the scale of a few years) driven by changes in large scale transport patterns in the troposphere. For our purposes, this sub-decadal variability is "noise"; we minimize the obscuring effects of this variability by selecting analysis techniques that effectively average over this variability.

### 2.1 Model Simulation Results


Time series of monthly mean ozone concentrations simulated by ESMs are our primary basis for investigating changes in the seasonal ozone cycle; an example time series is shown in Figure 1. These ozone time series come from six different CMIP6 ESMs: BCC-ESM1, CESM2-WACCM, GFDL-ESM4, GISS-E2-1-H, MRI-ESM2-0, and UKESM1-0-LL. Table S2 gives references for descriptions of these ESMs and their model output. Results of CMIP6 model simulations are archived at the

Earth System Grid Federation (https://esgf-node.llnl.gov/projects/cmip6/) and is freely available to download. We obtained monthly mean ozone concentrations for all six ESMs at the model levels that correspond to the selected comparison

locations. Where available, a mean of multi-ensemble members was calculated for each model from the CMIP6 historical simulations over the period 1850-2014.

### 2.2 Fit Equations


We fit time series of monthly mean ozone concentrations with the following equation, which has separate functions for the average long-term change (LTC) and the superimposed seasonal cycle (SC):

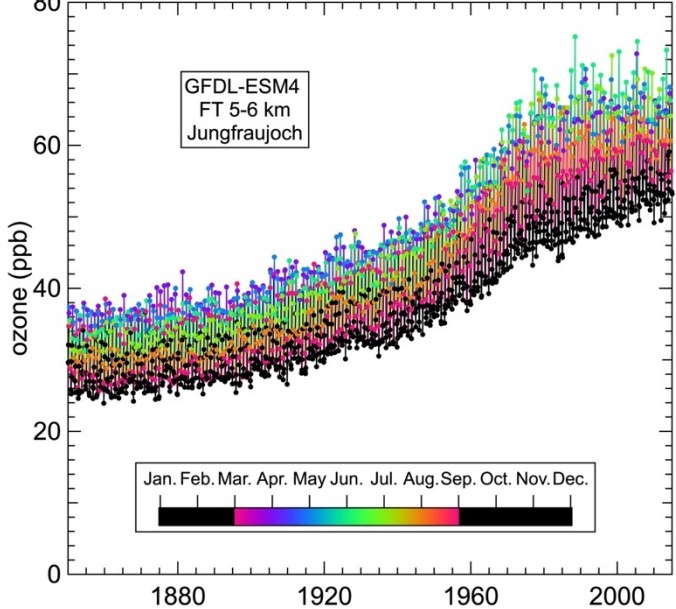

**Figure 1. Example time series of 165-years of monthly mean ozone concentrations simulated by the GFDL-ESM4 model in the FT between 5-6 km above Jungfraujoch. Ozone concentrations are colored according to month of the year to illustrate the phase shift of the seasonal cycle.**



$$O_3(t) = LTC(t) + SC(t), \qquad (1)$$

where t is time in years. We find representing LTC(t) by a 5-term power series (i.e., a 5-term polynomial):

$$LTC(t) = intercept + slope*t + curve*t^2 + d*t^3 + e*t^4, \qquad (2)$$

accurately quantifies the long-term change in average ozone concentrations, and allows the effective detrending of the monthly means for quantification of the seasonal cycle (see discussion in Section S1 of the Supplement). The choice of five
polynomial terms in the power series is somewhat arbitrary, with a range of polynomial terms (2 to 12) successfully detrending the monthly means without affecting the parameter values derived in the quantification of the seasonal cycle to a statistically significant extent. The influence of different power series fits on the derived seasonal cycle parameters is discussed in more detail in Section S1 of the Supplement.

Quantification of the seasonal cycle is complicated by significant shifts seen in both the amplitude and phase of the
seasonal cycle over the last ~75 years of the time series as illustrated in Figure 1. Annual maximum ozone moved from primarily March and April (pink, purple, or blue) before 1900 to primarily June (blue-green and light green) by the 1980s; concurrently, the vertical spread of the time series increased from ~12 ppb before 1900 to ~20 ppb by 1980. SC(t) must capture both the preindustrial seasonal cycle (dominant for the first ~90 years of the time series) and the later-occurring seasonal cycle shifts. A 2-term Fourier series quantifies the preindustrial seasonal cycle (PISC) in the detrended monthly
means:

$$PISC(t) = A_1*sin[2\pi*t + \varphi_1] + A_2*sin[4\pi*t + \varphi_2], \qquad (3)$$

where $A_1$ and $\varphi_1$ are the amplitude and phase, respectively, of the fundamental (one sine cycle year[-1]), and $A_2$ and $\varphi_2$ are the amplitude and phase, respectively, of the second harmonic (two sine cycles year[-1]) of the Fourier Series. As discussed in Section S2 of the Supplement, we find that the fundamental is larger in magnitude than the second harmonic, and together
the fundamental and second harmonic capture nearly all the variance associated with the seasonal cycle. Higher order harmonics are not included in Equation 3 due to their small magnitude compared to the fundamental and second harmonic.

The inclusion of two Gaussian functions serves to quantify the shifts in the seasonal cycle. These Gaussian functions are added to the $A_1$ and $\varphi_1$ parameters in the first term of the PISC(t) function to quantify changes in the amplitude and phase of the fundamental harmonic without affecting the values of the $A_1$ and $\varphi_1$ parameters. Once the Gaussian functions are
included, the seasonal cycle function is complete:

$$SC(t) = (A_1 + r*exp\{-((t-m)/s)^2\})*sin[2\pi*t + (\varphi_1 + r_\varphi*exp\{-((t-m_\varphi)/s_\varphi)^2\})] + A_2*sin[4\pi*t + \varphi_2]. \qquad (4)$$

This equation quantifies both the PISC(t) that is seen at the beginning of the ozone time series and the shift of the seasonal cycle later in the time series. In SC(t), the r and $r_\varphi$ parameters represent the magnitude of the Gaussian functions, the m and $m_\varphi$ parameters represent the time of their maximum values, and the s and $s_\varphi$ parameters represent their widths. Gaussian
functions are only included in the fundamental term of the Fourier Series because it is the only harmonic to consistently shift across locations and models. Separate Gaussian functions describe the shifts in the magnitude and phase of the seasonal cycle so that independent shifts of both components can be quantified. Note that in Equation 4 the $A_1$ and $\varphi_1$ parameters characterize the amplitude and phase of the preindustrial seasonal cycle, since the widths and the times of the maxima of the


Gaussians are such that they contribute negligibly in 1850; in comparing parameters derived here with parameters derived

from a similar equation that does not include the Gaussian terms (e.g., in Parrish et al., 2019), the sums $A_1 + r$ and $\varphi_1 + r_\varphi$

derived here are most appropriate to compare to the $A_1$ and $\varphi_1$ valued derived in that earlier work, which analyzed the ozone

seasonal cycle from observations collected in the last few decades.

Substitution of Equations 2 and 4 into Equation 1 gives a 7-term equation with 15 independent parameters:

$$O_3(t) = \text{intercept} + \text{slope}*t + \text{curve}*t^2 + d*t^3 + e*t^4 +$$

$$(A_1 + r*\exp\{-((t-m)/s)^2\})*\sin[2\pi*t + (\varphi_1 + r_\varphi*\exp\{-((t-m_\varphi)/s_\varphi)^2\})] + A_2*\sin[4\pi*t + \varphi_2]. \quad (5)$$

In the following analysis ozone time series are fit to this equation, which consistently captures more than 95% of the

variance in the time series of monthly mean ozone. Note that Equation 5 is parallel to Equation 4 of Parrish et al. (2019); the

greater complexity here is due to the inclusion of polynomial terms through 4[th] order in LTC(t) and the Gaussian terms

describing the shift of the seasonal cycle introduced in Equation 4. The longer monthly mean ozone time series from the

model simulations analyzed here allow the more complex Equation 5 to be fit with good statistical precision. In fitting

Equation 5 to a time series of monthly means, the derived parameter values are more precisely fit if the time origin is chosen

within the time series span. Here we choose year 2000 (i.e., t in above equations equals year-2000); Parrish et al. (2019) fully

discuss the implications of this choice.

We fit time series of the annual ozone precursor emissions (PE) that are prescribed in the models from anthropogenic and

biomass burning sources with an equation similar to that fit to the ozone time series; it has separate terms for the pre-

industrial long-term change (PITC) and more complex behavior (EG) during industrial development:

$$PE(t) = PITC(t) + EG(t). \quad (6)$$

A linear function:

$$PITC(t) = \text{intercept} + \text{slope}*t \quad (7)$$

accurately quantifies the early long-term change in average precursor emissions. The EG term of Equation 6 is given by a

Gaussian function that is consistent with an increase and then decrease in emissions primarily driven by anthropogenic

activity:

$$EG(t) = r_{em}*\exp(-((t-m_{em})/s_{em})^2) \quad (8)$$

This term is analogous to the Gaussian functions describing the shifts in the seasonal ozone cycle in Equations (4) and (5).

Here $r_{em}$ represents the maximum of the Gaussian, $m_{em}$ represents the year of that maximum, and $s_{em}$ represents its width.

Substitution of equations (7) and (8) into Equation (6) gives a 3-term fit equation with 5 parameters:

$$PE(t) = \text{intercept} + \text{slope}*t + r_{em} *\exp(-((t-m_{em})/s_{em})^2), \quad (9)$$

which captures more than 98% of the variance in the precursor emission time series.

Fitting of ozone and precursor emissions time series with these similar equations allows for quantitative comparison

between the ozone seasonal cycle shift and the growth and decrease in emissions. Specifically, comparison of the ozone

seasonal cycle Gaussian parameters in Equations (4) and (5) with the precursor emission Gaussian parameters in Equations





(8) and (9) is the basis for examining the correlation between changes in the seasonal cycle and changing ozone precursor emissions.

The multivariate regression fits of Equations 5 and 9 to time series of monthly mean ozone concentrations and annual emissions, respectively, quantify confidence limits for all derived parameter values. In this work 95% confidence limits are tabulated and discussed throughout. However, these confidence limits only reflect the variability of the time series about the functional form fit to that time series, and this approach assumes that each member of the time series is an independent variable with no autocorrelation within the time series; hence it must be recognized that these confidence limits are lower estimates of the actual uncertainties of the derived parameter values.

**2.3 Selected CMIP6 Simulation Locations**

The CMIP6 ESMs provide monthly mean ozone concentrations on global grids. To focus our investigation on northern midlatitudes, model-simulated monthly mean ozone time series are taken from model cells at three locations in western Europe and three in western North America. European locations are surface sites at Hohenpeissenberg, Germany and Jungfraujoch, Switzerland, and in the FT above Jungfraujoch at altitudes between 5 and 6 km. North American locations are

located in California - one surface site in the US at Lassen Volcanic N.P., and in the FT above Trinidad Head at two different altitudes: between 0.9 and 1.2 km in different models (we refer to this site as Trinidad Head at 1 km for simplicity), and between 5 and 6 km. Table 1 summarizes the location details including surface site elevations. In some cases, the model cells containing surface sites were not the lowest model cell; instead model cells with the average elevation closest to actual site level were chosen. For example, cell altitudes varied between 3.2 and 3.8 km for Jungfraujoch.

These six evaluation locations were chosen for three key reasons. First, there are measurement records at these locations spanning from two to nearly five decades, which allows for the quantification of the observed seasonal cycle and comparison between models and measurements. Second, the sites chosen on both continents have somewhat similar environments. Sites on both continents are in the western continental regions, which allows transported baseline ozone to dominate the ozone concentrations. Each continent includes a location in the FT between 5 and 6 km, an elevated surface site (Jungfraujoch and

Lassen Volcanic NP), and a location situated at or near the 1km elevation (Hohenpeissenberg and the FT above Trinidad Head at 1 km). The lowest elevation sites are a surface site in Europe and a sampling of the troposphere at 1 km altitude over North America, so there is not exact correspondence in site selection between the two continents. Finally, the sites chosen are representative of multiple different environments: low-, medium-, and high-altitude sites locations in both Europe and North America. The sites on each continent are all within ~500km; given the pronounced zonal similarity of ozone concentrations at

midlatitudes (Parrish et al., 2020), the geographic separation between these sites has negligible impact on ozone concentrations, so the two sets of three sites are representative of their respective continents at different altitudes or elevations. In summary, the locations are selected to provide an altitude-dependent contrast between the western regions of the two continents.



**Table 1. Locations included in seasonal cycle analysis.**

| Site | Latitude/Longitude | Model Simulations or Measurements | Surface Site or Sondes | Elevation (km) |
|---|---|---|---|---|
| Hohenpeissenberg, Germany[a] | 47°48′N/11°1′E | Both | Both | 0.98 (surface) 5-6 (sonde) |
| Jungfraujoch, Switzerland[b] | 46°33′N/7°59′E | Both | Both | 3.6 (surface) 5-6 (sonde) |
| Zugspitze, Germany[b] | 47°25′N/10°59′E | Measurements | Surface site | 3.0 |
| Sonnblick, Austria[b] | 47°3′N/12°57′E | Measurements | Surface site | 3.1 |
| Uccle, Belgium[a] | 50°48′N/4°21′E | Measurements | Sondes | 5-6 |
| Payerne, Switzerland[a] | 46°49′N/6°57′E | Measurements | Sondes | 5-6 |
| Trinidad Head, US | 41°3′N/124°9′W | Both | Sondes | 0.9-1.2, 5-6 |
| 200 km West of Trinidad Head, US[c] | 41°3′N/126°30′W | Model Simulations | Sondes | 0.9-1.2, 5-6 |
| Lassen Volcanic N.P., US | 40°32′N/121°35′W | Both | Surface site | 1.8 |

[a] Sonde measurements from Hohenpeissenberg, Uccle, and Payerne are averaged to form the European FT data set, more detail given in Section 2.4

[b] Surface measurements from Jungfraujoch, Zugspitze, and Sonnblick are averaged to form the European Alpine data set, more detail given in Section 2.4

[c] Offshore location selected for comparison with onshore; details included in Section S4

**2.4 Ozone Observations**

Although model simulations are our main basis for analysis, observational data are also considered. Shifts in the amplitude and phase of the seasonal ozone cycle are generally apparent in observational records that span long enough time periods. The measurements serve to check the accuracy of model simulations; realistic model simulations are expected to at least

approximately reproduce the observed seasonal cycle and its temporal shifts.

Our analysis includes three observational data sets from both Europe and North America. The European data sets include one spanning 47 years, 1971-2016, at Hohenpeissenberg, Germany, one spanning 40 years, 1978-2017, averaged between three European alpine sites: Jungfraujoch, Switzerland; Zugspitse, Germany; and Sonnblick, Austria; and one spanning 20 years, 1998-2017, averaged between measurements from sondes launched from European ground sites at Hohenpeissenberg,

Germany; Uccle, Belgium; and Payerne, Switzerland. We consider average sonde measurements between 5 and 6 km. The impetus behind averaging measurements from three surface sites and three sonde data sets is to reduce the impact of ozone variability in any one data set, and to thereby obtain a more precise quantification of ozone over Western Europe. These same data sets have been considered in previous studies of western European baseline ozone concentrations. The Hohenpeissenberg data discussed by Parrish et al. (2014) are here extended through 2016, and the European alpine and

sonde data sets are the same as those analyzed by Parrish et al. (2020). The North American data sets include one spanning



30 years, 1998-2017, at Lassen Volcanic N.P., and one spanning ~ 21 years, late 1997-early 2018, from sondes launched from Trinidad Head. We consider average sonde measurements between 5 and 6 km, and between 0.5 and 1.0 km. These North American data sets are also the same as those analyzed by Parrish et al. (2020).

## 2.5 Precursor Emissions

Annual mean ozone precursor emissions were derived from ESM emission inventories integrated over the northern midlatitude region between 30° and 60° N for the 1850-2014 simulation period. The primary analysis examines emissions of $NO_X$ and VOCs from anthropogenic (Hoesly et al., 2018) and biomass burning sources (van Marle et al., 2017) that were provided as a common emission inventory to be used by all models (including the six in this study) in CMIP6 simulations. As discussed in further detail in Section S5 of the Supplement, the anthropogenic emissions dominate this inventory.

Although there are small seasonal cycles in these emissions, these seasonal cycles are either approximately constant over the entire time interval, or their relative magnitudes are small compared to that of the seasonal cycle of ozone; further discussion is included in Section S5.

Even though the six ESMs used the same prescribed anthropogenic and biomass burning emissions, Figure 1 of Griffiths et al. (2021) shows that subtle differences remain in NOx emissions and even greater differences in CO and biogenic VOC

emissions between models. Differences in the VOC emissions arise because the speciated VOC emissions that were provided had to be mapped onto the chemical mechanisms in the individual models, and this mapping may not fully account for the total VOC emissions prescribed. The emissions that are the focus of our analysis have been taken from the prescribed emission inventory; we have not further diagnosed the exact northern midlatitude emissions actually used in each individual model.

Some of the models were able to provide quantifications of emissions from biogenic and other natural sources for evaluation. These emissions varied between models based on model-specific chemistry and parameterizations, and included biogenic VOC emissions (specifically, isoprene) and dimethyl sulfide (DMS) from oceans, and $NO_X$ emissions from soil and lightning. Methane is considered independently due to its very long lifetime compared to other VOCs; all of the ESMs use prescribed global annual mean values of $CH_4$ concentrations as input at the surface throughout the whole historical period

(Meinshausen et al., 2017). Further details of model-specific natural emissions and $CH_4$ are given in Section S6 of the Supplement.

## 3 Results

### 3.1 Isolating the seasonal cycle: Detrending monthly means and harmonic analysis

Ozone in the troposphere varies on a wide spectrum of temporal scales, which makes it difficult to quantify a particular

contribution to that variability. To isolate the seasonal cycle, we examine time series of monthly mean ozone concentrations.





Monthly means integrate over the short-term variability driven by diurnal cycles and short-term meteorological changes, which effectively removes their influence. The time series considered here span a maximum of 165 years, which allows significant influence from "longer-term" (i.e., on the scale of decades to centuries) variations driven by ozone precursor emission changes and climate variations. We isolate the seasonal cycle from these longer-term changes by detrending the

monthly mean concentrations, which we accomplish by subtracting LTC(t) in Equation 1 from the time series of monthly means. A regression fit of the five-term polynomial given in Equation 2 to the time series of monthly means gives values for the five polynomial coefficients; Section S1 of the Supplement discusses more details of the determination of LTC(t). Figure 2 shows the example time series of Figure 1 with the fit to Equation 2 indicated by the black curve, which quantifies the longer-term temporal change that underlies the time series of monthly means. Figure 2 also includes a fit to the complete

Equation 5, shown in red. The seasonal cycle of the detrended monthly means is apparent as variation of the monthly means (blue dots) about the black curve. As expected, the detrended monthly means display an annually repeating seasonal cycle.

Any repeating signal, such as the seasonal ozone cycle, can be quantified by a linear combination of sinusoidal functions (i.e., a Fourier series). The seasonal cycles we examine are described sufficiently by the sum of the first two harmonics: the fundamental (one sine cycle yr$^{-1}$) and the second harmonic (two sine cycles yr$^{-1}$) as indicated in Equation 3. The fundamental

is generally larger in magnitude than the second harmonic, except for the two lower-elevation North American sites, for which the two harmonics were approximately equal in magnitude during the preindustrial period. In combination, the fundamental and second harmonic capture almost all the variance associated with the seasonal cycle. A quantitative Fourier analysis that provides the basis for this harmonic analysis and the inclusion of only the first two harmonics is detailed in Section S2 of the Supplement. The detrended seasonal cycles, including their evolution over the course of the 1850-2014

period, are analyzed for the six ESM simulations at the six selected northern midlatitude locations; this is the primary basis of our analysis, which is discussed in the next three subsections.

### 3.2 Model-simulated preindustrial seasonal cycle

To understand the magnitude and timing of changes to the

seasonal ozone cycle that began near the middle of the 20th Century, it is important to quantify the seasonal cycle before those changes began, i.e., the preindustrial seasonal cycle. Only very limited ozone measurements are available before the mid-20th Century, so our quantitative analysis of

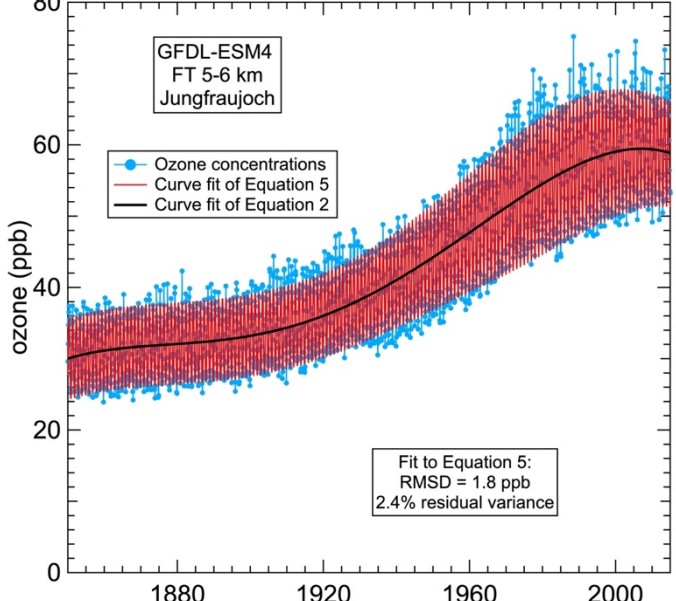

**Figure 2. Blue points indicate the same example time series of monthly mean ozone concentrations shown in Figure 1. The black and red curves indicate the fits of the 5-parameter, long-term change in Equation 2, and the full 15-parameter Equation 5, respectively.**



the preindustrial seasonal cycle is limited to model simulations. Fits of Equation 3 to the time series of detrended monthly means quantify the contributions of the fundamental and second harmonic; detailed descriptions of similar fits to time series of monthly means from observations in the MBL and FT are given by Parrish et al. (2016) and Parrish et al. (2020), respectively. Each fit provides 4 parameter values that quantify the preindustrial seasonal cycle for a model simulation at a particular location. Figure 3 quantitatively examines the simulated preindustrial seasonal cycle in the FT between 5-6 km at

locations above Europe and North America. Because these are higher-altitude locations, they are more physically separated from ground-based sources of emissions than are surface sites. We assume these locations are representative of the FT baseline seasonal ozone cycle with little influence from local or regional emissions; thus, they are appropriate for our initial analysis. At both FT locations, the preindustrial seasonal cycle is similar in character; it is determined largely by the fundamental, which generally reaches its seasonal maximum in May or June. Figures S4 and S5 give plots similar to Figure

3 for the four lower-elevation locations, with discussion in Section S3 of the Supplement.

Between different models, there are important qualitative similarities in the simulated preindustrial seasonal cycle at most locations. First, the fundamental is larger in magnitude than higher-order frequencies, except for Trinidad Head at 1 km and Lassen NP. Second,

the maximum of the fundamental occurs in the late spring or early summer, which drives an overall seasonal maximum that also occurs in the late spring or early summer. Marenco et al. (1994) report a similar seasonal cycle with a springtime

maximum based on late-19th Century observations at Pic du Midi, a remote mountaintop site in France; given the paucity of measurements from the preindustrial period, this is the strongest comparison available between measurements and

model simulations.

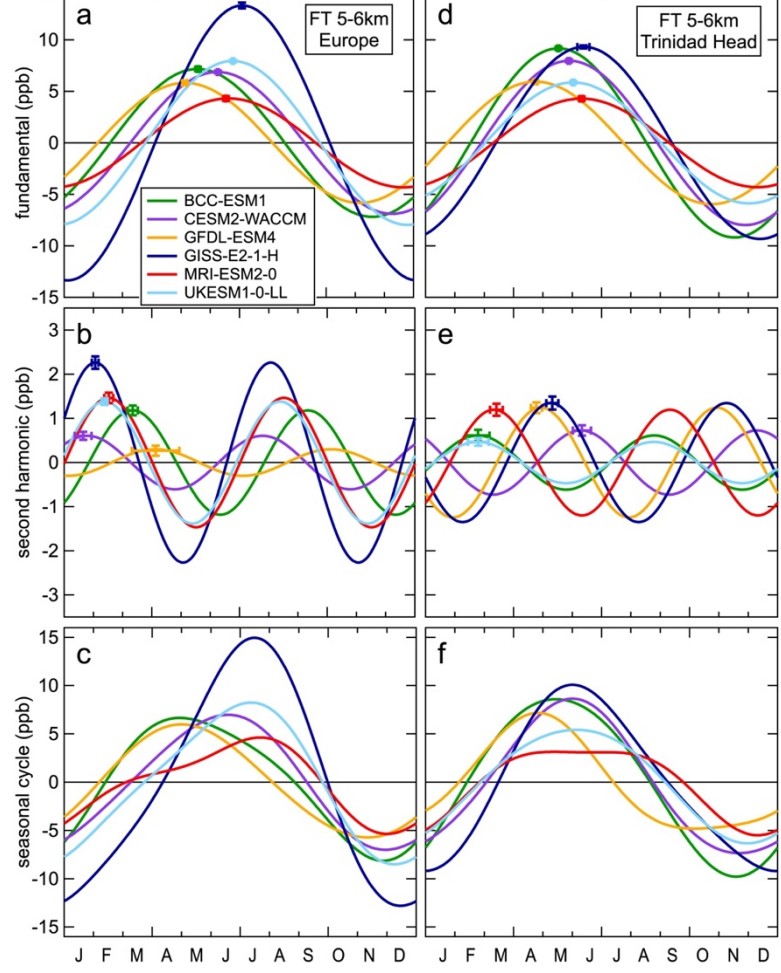

**Figure 3. Harmonic analysis of preindustrial seasonal cycle at two remote FT sites. Curves, color-coded according to model, give the fundamental (a and d), second harmonic (b and e) and total seasonal cycle (c and f), which is calculated from the sum of the two harmonics. Error bars at the maxima of the harmonic curves indicate the confidence limits of the amplitudes and phases (some are too small to clearly discern).**





Despite qualitative similarities, there are quantitative differences in simulations among models at specific sites, and within individual model results across different sites. Figure 3 indicates that the amplitudes of the simulated seasonal cycles vary by a factor of ~3. Exclusion of the GISS-E2-1-H model, which Griffiths et al. (2020) note simulates the strongest response of tropospheric ozone to precursor emissions of CMIP6 models, lowers this factor to ~2. Additionally, the models do not all

reproduce the degree of zonal similarity of the seasonal cycle at northern midlatitudes noted by Parrish et al. (2020); the amplitude and phase of both harmonics and the overall seasonal cycle differ significantly between the European and North American FT sites in some model simulations. These patterns are also present at the other, lower-elevation locations examined (Figures S4 and S5).

**3.3 Seasonal cycle shifts across northern midlatitudes**

Across all models and all locations, shifts in both the amplitude and phase of the seasonal cycle are ubiquitous. Importantly, the presence of a seasonal cycle shift in the FT indicates it is a hemisphere-wide phenomenon, rather than limited to a localized environment. Figure 4 compares preindustrial and modern-day seasonal cycle simulations

from one example model with the observed modern-day seasonal cycle in the FT over Europe. This is the same example time series shown in Figures 1 through 3. The modern-day seasonal cycle is larger in amplitude with a later maximum compared to the preindustrial seasonal cycle. These changes are primarily driven by

the changing fundamental, rather than the second harmonic, which makes only a small contribution in the FT and is not statistically different between the preindustrial and modern-day simulations. The modern-day simulated seasonal cycle approximates, but does not exactly match observations from the past two decades; the simulated

seasonal cycle is smaller in amplitude with an earlier maximum than the measured seasonal cycle.

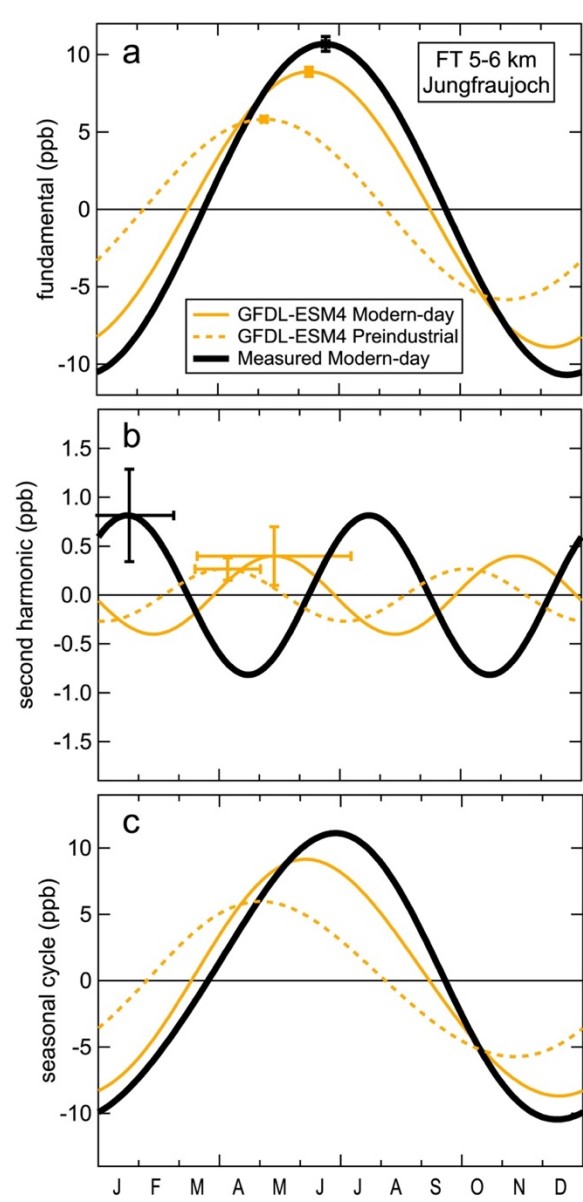

**Figure 4. Comparison of simulated preindustrial and modern-day seasonal cycles in the FT between 5-6 km above Jungfraujoch; the observed modern-day seasonal cycle is also included. The GFDL-ESM4 simulations are the same as shown in Figures 1 and 2. The preindustrial seasonal cycle is the same as included in Figure 3 and the format is the same as that figure: (a) shows the fundamental frequency, (b) shows the second harmonic, and (c) shows the sum of the two harmonics. The modern-day seasonal cycle was calculated over the 1985-2014 period, and the measured seasonal cycle is based on the 1998-2017 European sonde measurements.**



The temporal evolution of the fundamental harmonic for all model simulations and measurements is shown in Figures 5 and 6 for all six locations. The colored curves in these figures are derived from the fits of Equation 5 to the respective time series of simulated monthly means; they represent the evolution of the amplitude (left panels) and phase (right panels) of the

fundamental over the period of the simulations. In general, in each simulation the fundamental is approximately constant in magnitude and phase for approximately the first half of the time series, depending on model and site, before significant shifts begin. Most of the models agree that near the middle of the 20th century, the amplitude began to increase and the phase to change so that the seasonal maximum appeared later in the year compared to the preindustrial values. Near the end of the 20th century, these changes began to reverse. The Gaussian functions incorporated in Equations 4 and 5 are generally defined

precisely in the model simulation fits. However, fits to some simulated time series return no statistically significant parameters for the corresponding Gaussian function, and the resulting curve in Figure 5 or 6 is then a horizontal line; the MRI-ESM2-0 simulation at both FT locations is such an example. Such horizontal lines indicate either that the model simulated a constant fundamental amplitude or phase (i.e., no shift in that harmonic property), or that the variability in the simulated monthly means was

too large to allow a statistically significant measure of the shift in the fundamental phase or amplitude to be discerned.

**Figure 5. Shift of the seasonal ozone cycle at the three European locations represented by changes in parameter values fit by the Gaussian functions of Equation 4. Left and right panels quantify the amplitude and phase, respectively, of the fundamental as a function of year. The left axes in the right panels give the date of the seasonal maximum, while the right axes show corresponding values of the phase in radians. Dashed lines extend from the maximum value of the Gaussian function to the x-axis, indicating the $m$ and $m_\varphi$ parameters; associated error bars indicate confidence limits of the parameters derived from the measurements. Colors identify the respective model simulations. From bottom to top, the panel positions correspond to relative elevations.**

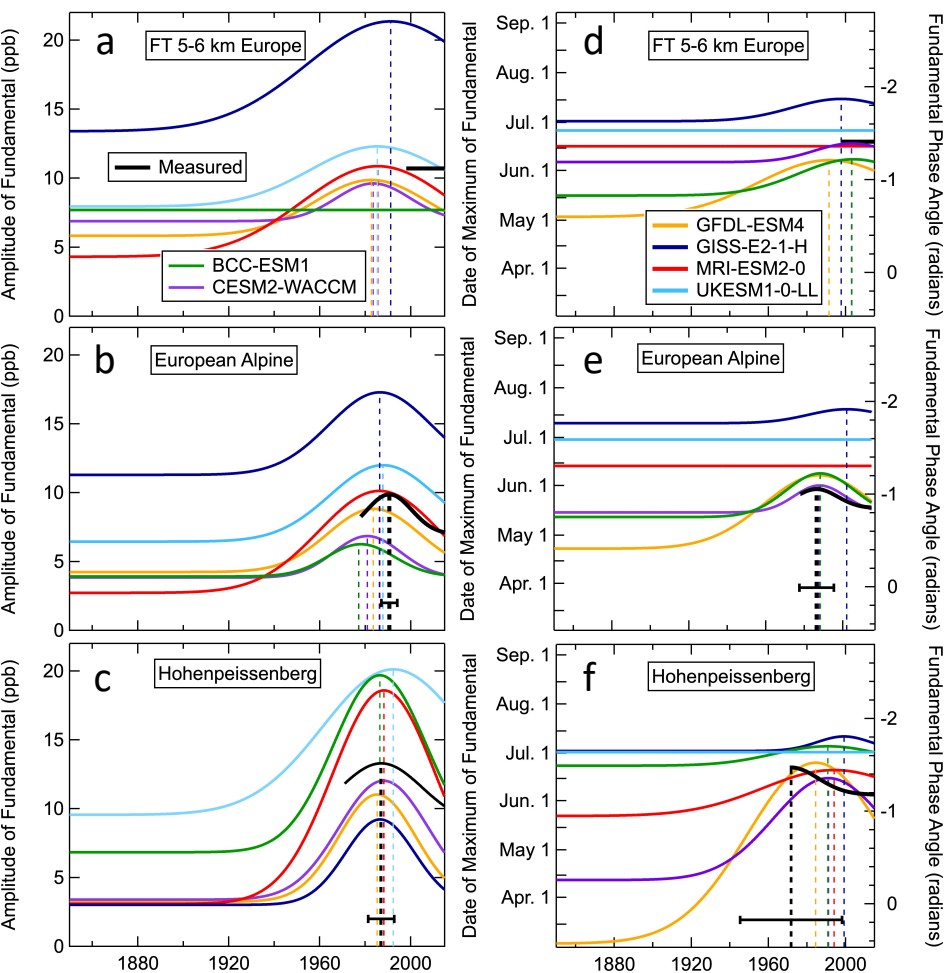






The fits of Equation 5 to the measured time series (black curves) are much less certain than the fits to the model

simulations due to the much shorter period of the measurements, which also generally exhibit greater variability. In all of the

measured time series from the FT and at some surface sites, only the average of the amplitude and phase of the fundamental

over the measurement period can be extracted from the available data; in Figures 5 and 6 these averages are indicated by

horizontal line segments that span that measurement period. The two longest measurement records were collected at the two

European surface sites; Figure 5 shows the shifts in the seasonal cycle extracted from these records. In North America, only

the seasonal cycle phase at Lassen Volcanic NP (Figure 6e) shows a significant shift; however, that shift can only be

quantified by a linear, one-parameter function (equal to the slope) replacing the Gaussian, three parameter function in

Equation 5; this fit is indicated by the sloping line segment in the figure that spans the measurement period. This line

segment does approximate the shape of the Gaussian fits to two of the corresponding model simulations. It should also be

noted that the linear fit to the phase shift is closely related to an earlier analysis approach (Parrish et al., 2013) that also

quantified the phase shifts from
observations at some of these
same sites. Table 2 compares the
present results with those earlier
ones. Overall, the results agree
within their confidence limits at
Hohenpeissenberg, the European
alpine sites (Jungfraujoch and
Zugspitze analyzed separately in
the earlier work), and Lassen
Volcanic NP. At the European
sites the present results do
indicate smaller slopes, which is

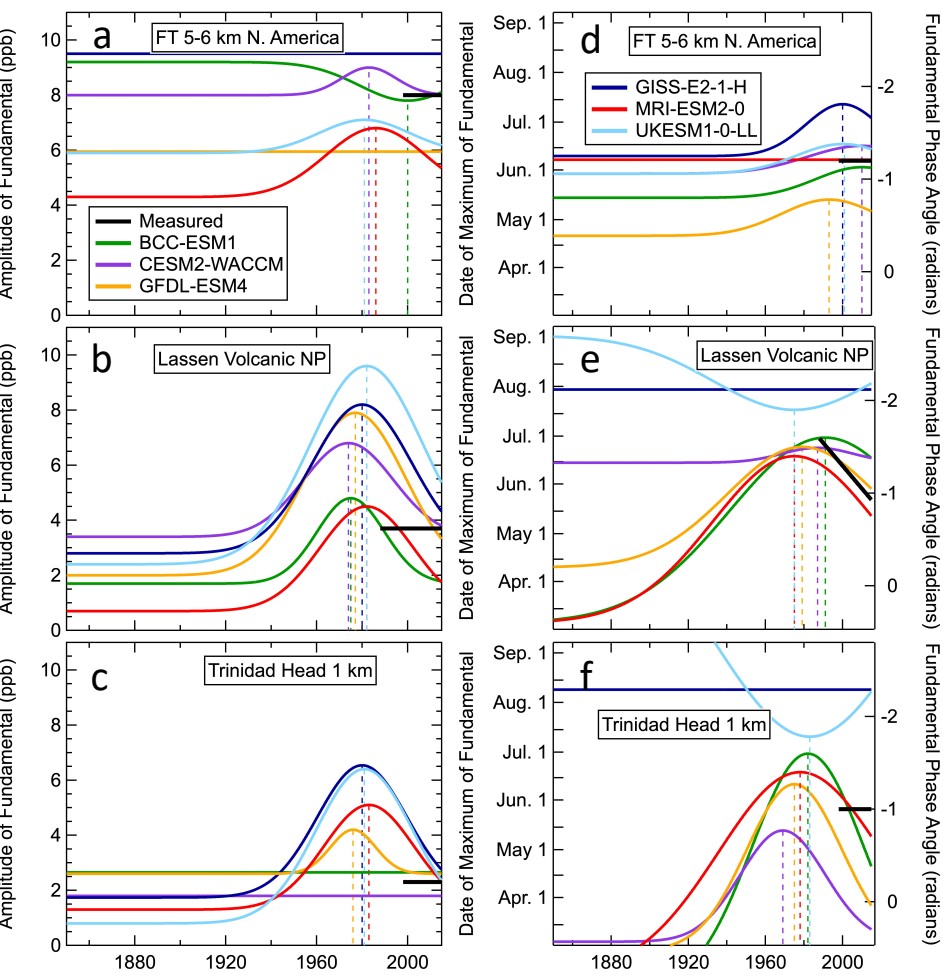

**Figure 6. Same as Figure 5, except for the three North American locations, with expanded ordinate scales on the left panels. The simulated phase of the fundamental at Trinidad Head at 1 km goes off-scale for most simulations at some point during the time series; Figure S6 of the Supplement shows the Trinidad Head phase dependence on an expanded scale.**





consistent with their inclusion of data from more recent years when the shift in the phase of the seasonal cycle was slowing.

**Table 2. Linear fits to shifts in the phase of ozone seasonal cycle analysis; units are days/decade.**

| Site | Parrish et al., 2013 | This work |
|---|---|---|
| Hohenpeissenberg, Germany | 6.4 ± 2.4 (1971–2010) | 4.5 ± 1.9 (1971–2016) |
| Jungfraujoch, Switzerland | 5.6 ± 4.1 (1990–2010) | --- |
| Zugspitze, Germany | 5.1 ± 3.5 (1978–2009) | --- |
| European alpine sites | --- | 3.7 ± 2.5 (1978-2017) |
| Lassen Volcanic N.P., US | 14 ± 19 (1988–2011) | 14 ± 9 (1988–2017) |

**3.4 Connection between ozone precursor emissions and the seasonal cycle**

     All six CMIP6 ESM simulations incorporated the same ozone precursor emission inventory for anthropogenic and biomass burning sources. Figure 7a illustrates the temporal evolution of these nonmethane VOC and NOx emissions

integrated annually and over the entire northern midlatitude region (30° to 60° N). The curves in Figure 7b are fits of Equation 9 to those emissions; these fits (with the underlying linear increases) capture more than 98% of the variance in the time series of annual emissions. Equation 9 is designed to provide Gaussian function fits to the emissions, so that the derived parameters can be directly compared to the Gaussian parameters that

quantify the shift of the ozone seasonal cycle. Figures S10 and S11 compare the Gaussian parameters from the emission fits with those derived from fits to the model simulated ozone at individual sites.

     The parameters from individual model simulations exhibit large variability between the six locations, particularly at lower elevations.

To more precisely compare the seasonal cycle shifts with the temporal evolution of the emissions, we average the Gaussian parameters in various ways over sites and model simulations. We average over all three European locations, all three North American

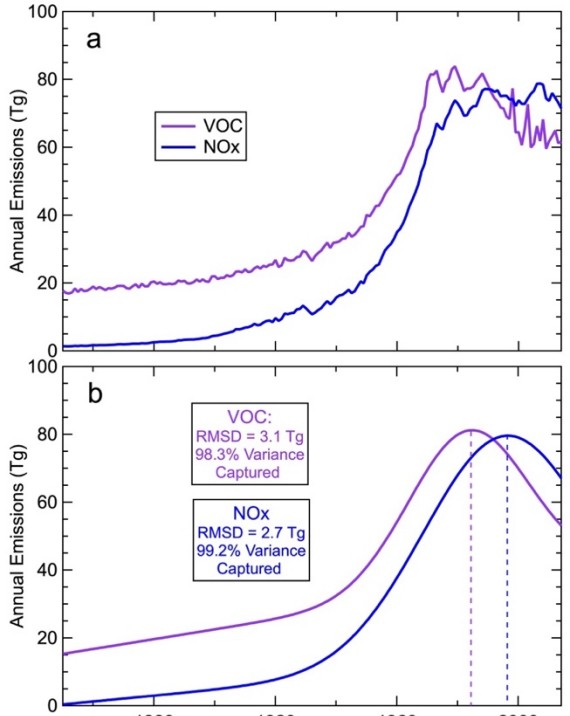

**Figure 7. Ozone precursor emissions from anthropogenic and biomass burning sources, which are common in all six CMIP6 ESMs, integrated over northern midlatitudes. (a) Annual emissions and (b) fits of Equation (9) to those emissions with fit statistics annotated. Vertical dashed lines indicate the year of the maximum emissions ($m_{em}$ parameter) as in Figures 5 and 6.**





locations, the three higher elevation locations (the FT above Trinidad Head and Jungfraujoch between 5-6 km, and the
European alpine sites), the three lower-elevation sites (Hohenpeissenberg, Lassen Volcanic N.P., and the troposphere above
Trinidad Head at 1 km), each model simulation at all six sites, and an overall average over all six simulations at all six
locations. Figure 8 compares these averages with the Gaussian parameters from the emission fits and the limited
measurement results; these averages are also included in Figures S10 and S11. Tables 3 and S3 list some of these model
simulation averages, along with the ozone precursor emission and measurement parameters. The averaging of parameters
across different selections of simulations and locations minimizes the influence of localized emissions and any site-specific
behavior. All of these results are weighted averages, where each parameter value from an individual simulation result is
weighted by the inverse of the square of the confidence limit of that parameter. In Figure 8 the parameters derived at the six locations by the individual

**Figure 8. Summary of ozone seasonal cycle shift analysis. The left and right graphs illustrate the shifts in the amplitude and phase of the seasonal cycle, respectively, for the year of maximum shift (a and b), half-width of the Gaussian function fit to the shifts (c and d) and the maxima of the shifts (e and f). Circles indicate weighted averages of the parameters derived in all fits: blue filled circle is the average for all six models at all six locations, the six-location averages for each model are to the right, and six-model averages for three locations selected for continent or elevation are to the left. Parameters derived from fits to observations at the European alpine sites (EAS) and Hohenpeissenberg (HPB) and VOC and NOₓ emissions are included near the center of each graph in the annotated symbols. Error bars indicate confidence limits for all symbols, although many are covered by the symbols themselves.**



**Table 3. Gaussian parameters that define changing emissions and seasonal ozone cycle shifts over northern midlatitudes. First two rows give fit parameters for total anthropogenic and biomass burning ozone precursor emissions integrated across the entire northern midlatitude region (30° to 60° N); second two rows give parameters for fits to observed and model simulated seasonal cycles. Positive *r* values for the phase shift indicate a seasonal cycle shifting towards a later annual maximum. The seasonal cycle Gaussian parameters are averaged over the six locations considered in the analysis.**

| Northern midlatitudes | Gaussian maximum, *m* parameter (Year) | | Gaussian amplitude, *r* parameter | | Gaussian amplitude, *r* parameter | |
|---|---|---|---|---|---|---|
| | Phase | Amplitude | Phase (days) | Amplitude (ppb) | Phase | Amplitude |
| NO$_X$ Emissions | --- | 1995 ± 1 | --- | 67 ± 3 Tg | --- | 38 ± 2 |
| VOC Emissions | --- | 1983 ± 1 | --- | 47 ± 3 Tg | --- | 28 ± 2 |
| Simulations[a] | 1985.2 ± 0.5 (2.6) | 1985.2 ± 0.3 (1.3) | 22 ± 0.6 (10) | 5.4 ± 0.1 (0.6) | 39 ± 1 (2.5) | 29.5 ± 0.4 (1.6) |
| Observations[b] | 1985 ± 8 | 1990 ± 3 | 14 ± 8 | 2.9 ± 1.4 | 17 ± 17 | 15 ± 9 |

[a] Weighted mean over all six model simulations at all six sites; numbers in parentheses are estimated upper limits of confidence limits

[b] Includes results from Hohenpeissenberg and European alpine sites only

models are omitted for clarity; Figures S10 and S11 show those same graphs with the individual model/location parameters included with their confidence limits. These figures serve to collect the results of the analyses, and provide the basis for discussion of these results in the following section.

Interpretation of the confidence limits quoted for the derived parameters is difficult. The multivariate regressions utilized to fit the model simulations, observations and emissions return parameter values with 95% confidence limits, which are plotted in Figures 3, 4 and 8; many are not visible because they are smaller than the plotted symbols. These confidence limits are underestimated (see Section 2.2) due to autocorrelation in the time series of monthly mean ozone concentrations. An independent estimate of the confidence limit of each overall average parameter value can be obtained from the variance of the individual parameter values included in the average. If one assumes that each seasonal shift parameter must be identical at all six locations, and that each model simulation at each site provides an independent determination of that parameter value, then the confidence limit of the average can be estimated from the square root of the variance divided by the number of independent model determinations (36 if the fits to each of the six model simulations returns a parameter value at each of six locations). Such upper limits are included in parentheses in the bottom line of Table 3 and are included as the blue error bars on the overall averages in Figure 8; they are larger by factors of 2.5 to 17 than those derived from the weighted averages of the parameters from the regression fits. In quantitative comparisons of the parameters from observations and emissions with those simulated, this issue with the confidence limits must be considered.

## 4 Discussion and Conclusions

We analyze the seasonal cycle of tropospheric ozone over the historical period, as simulated by six CMIP6 Earth System Models and deduced from available observations at six northern midlatitude locations in western Europe and western North America. Over the time period of the model simulations (1850-2014), the seasonal cycles shifted significantly in both phase





and amplitude at all locations, including within the free troposphere. The seasonal cycles simulated by the models remained

generally constant from 1850 until well into the 20th century; this preindustrial seasonal cycle is shown in Figure 3 for two
       FT locations and in Figures S4 and S5 for four lower-elevation locations. In the period from approximately 1920-1940 the
       seasonal cycle amplitude began to increase, and the seasonal maximum began to shift to later in the year. These changes
       reached their maximum extent late in the 20th Century, after which they began to reverse - the seasonal cycle decreased in
       amplitude and the annual maximum shifted back to earlier in the year. Gaussian functional fits quantify these shifts.

Observations are available for at most only the last 44 years of the model simulations; within their large uncertainties (see
       error bars in Figure 8) the available measurements indicate seasonal cycle shifts similar to those simulated. Figure 4
       illustrates these shifts as simulated by one model at one location; it shows that the fundamental harmonic is the primary
       contributor to both the seasonal cycle and its shifts. Figure 4 also compares the simulated modern-day seasonal cycle with
       that derived from observations. Figures 5 and 6 show comparisons of the shifting amplitude and phase of the fundamental

harmonic among all models and with available observations at the six locations considered. The Introduction discussed
       extensive literature reports of modelled and observed changes in the seasonal ozone cycle throughout northern midlatitudes
       over the most recent three to four decades; the seasonal cycle shifts examined here are generally consistent with those
       reports.

       Throughout northern midlatitudes, on average (blue symbols in Figure 8) the simulated shifts in both the amplitude and

phase of the fundamental of the seasonal cycle maximize at similar times (~1985; Figures 8a, b) with the amplitude shift
       having a somewhat smaller width (~30 years; Figure 8c) than the phase shift (~40 years; Figure 8d). At the maxima, the
       fundamental amplitude (Figure 8e) had increased by ~5.5 ppb (i.e., a ~11 ppb increase in the difference between the seasonal
       minimum and maximum), and the seasonal maximum (Figure 8f) had shifted to ~3 weeks later in the year. For comparison,
       the average simulated preindustrial seasonal cycle in the free troposphere had an amplitude of ~7 ppb and a seasonal peak

near June 1 (Figures 5 and 6). The sparse measurement record from the European alpine sites and Hohenpeissenberg (red
       and green points in Figure 8 and entries in Table 3) agrees well for the timing of the maximum shifts, but suggests somewhat
       smaller seasonal cycle changes in the widths and magnitudes of those shifts; however, the large uncertainty in the
       observational determinations should be noted.

       The model simulations exhibit large variability, both among models and locations (compare points on right side of graphs

in Figures 8, S10 and S11); however, it is difficult to judge if this variability is statistically significant. Here we identify
       some aspects of this variability that appear to be robust. First, the relative spread among the model averages in the phase
       shift is greater than that in the amplitude for all three parameters (year of maximum, and the width and magnitude of the
       Gaussians quantifying the shifts). Second, both the amplitude and phase shifts appear larger and more varied at lower
       elevations compared to the FT (compare lower graphs in Figures 5 and 6 with the FT results in the upper graphs, and the low

and high elevation averages in Figures 8e,f and S12); since the anthropogenic emissions are located at the surface, this
       behavior may reflect the greater influence from local and regional emissions at the surface sites compared to the more
       isolated locations in the FT. Third, Hohenpeissenberg (located at a relatively low elevation in central Western Europe)


generally shows the largest amplitude shifts in the model simulations as well as in the measurements, although the measurement results are highly uncertain. At Hohenpeissenberg (Figure 5c) all six models simulated the timing of the

maximum amplitude shift (i.e., the *m* parameter) within the uncertainty of that derived from the measurements ($1987 \pm 6$ years). This temporal agreement occurs despite disagreement (by a factor of ~2) in the maximum fundamental amplitude (peaks of Gaussian curves in the Figure 5c) and disagreement (up to a factor of ~3) by 2 of the 6 models in the amplitude of the preindustrial fundamental (horizontal portion of the curves at the left of figure 5c). There is poorer agreement regarding the phase shift at Hohenpeissenberg, with the simulated maxima occurring between 1984 and 2000 in the six model

simulations; the timing of the maximum phase shift derived from the measurements is not precisely defined, but its confidence limits include (nearly) all of the model results. Fourth, the greatest variability of the simulated phase shifts is seen at Trinidad Head at 1 km (Figure 6f), which is the lowest elevation North American location considered here; there the maximum of the fundamental is found to occur in nearly every month of the year over the simulation period in at least one of the model simulations, although the seasonal cycle amplitude is relatively small at this location, which makes determination

of that maximum difficult. A possible explanation for these divergent model results is that this location is on the edge of two transitions – the MBL to FT and the marine to continental environment - which may be a particularly difficult situation for the models, which have coarse horizontal resolution, to simulate.

    We also quantify the temporal changes in total northern midlatitude ozone precursor emissions from anthropogenic and biomass burning sources (Figure 7) that are incorporated into the emission inventories assumed by all of the ESMs. Between

1850 and 1940 emissions increased only slowly, with more rapid increases beginning in the mid-20[th] Century as the result of rapid industrialization in Europe and North America. By the late 20[th] Century, emissions began to decrease as the result of air quality control efforts in more developed countries. The Gaussian fit to the $NO_X$ emissions in Figure 7 indicates a recent decrease, while the inventory shows approximately constant emissions; this is an artifact of representing these emissions with a Gaussian that is symmetric about the maximum. Notably, the changing emissions are driven by anthropogenic

activity; Section S5 of the Supplement compares the temporal changes of anthropogenic and biomass burning emissions, and shows that it is only the anthropogenic emissions that rise and fall over time, while the biomass burning emissions remain approximately constant.

    On average, the parameters that quantify the shifts in the seasonal cycle correlate strongly with those that quantify the emissions. Figures 8 a-d show that the overall model simulation averages of the four parameters that quantify the timing of

the shift of the amplitude and phase of the fundamental harmonic closely correspond to the parameters that quantify the temporal evolution of the emissions. There is significant variability in the results from the different models (open circles on the right in the graphs in Figure 8), but that variability is reduced in four regional averages (open circles on the left). There is no consistent, strong difference between the European and North American continents. Both the amplitude and phase shifts apparently maximized earlier in North America than Europe, but there is a great deal of variability among the individual

determinations (Figures S10 and S11) so the statistical significance of this apparent difference is uncertain. There also may be significant differences in the shapes of the Gaussian describing the phase shift between the lower-elevation surface sites


(i.e., earlier years of maximum shift and greater widths) compared to the higher-elevation sites representative of the FT (Figures 8b,d); and the phase shift at high elevations appears to have maximized later with a smaller width. The maxima of the amplitude and phase shifts (Figures 8e,f and S11) are apparently larger at the low elevation sites, which may reflect more

direct impact by anthropogenic emissions.

Based on the temporal correlation between the emission changes and the seasonal cycle shifts shown in Figure 8, we hypothesize that the changing ozone precursor emissions is the cause of the shifts in the seasonal ozone cycle throughout northern midlatitudes. During industrial development, ozone production driven by rising anthropogenic precursor emissions progressively becomes the predominant source of ozone, which shifts the ozone seasonal maximum into the summer, when

photochemical ozone production is more important (compared to, e.g., ozone input from stratospheric intrusions, which peaks in the spring). Ozone production driven by anthropogenic activity also increases the amplitude of the seasonal cycle by boosting summertime concentrations while wintertime concentrations are less affected. As emissions decrease, those changes reverse, with the seasonal cycle returning toward the pre-industrial cycle. Although ozone precursor emissions from all sources influence ozone production and the ozone seasonal cycle, it is anthropogenic activity that drives the seasonal

cycle changes; more discussion of natural and anthropogenic emissions is given sections S5 and S6 and Figure S7–S9 of the Supplement. The temporal correlation between the changes in emissions and the ozone seasonal cycle does not necessarily prove our hypothesis. Examination of some of the Aerosols and Chemistry Model Intercomparison Project (AerChemMIP; Collins et al., 2017) historical sensitivity experiments, where different drivers were fixed at pre-industrial levels may be suitable for a more definitive attribution of the causes of the ozone seasonal cycle shifts.

An interesting aspect of the correlation between precursor emissions and the ozone seasonal cycle shifts is the temporal offset in the evolution of the emissions. The Gaussian functions fit to the non-methane VOC emissions in Figure 7 (see Table S3a for parameter values) peaked in ~1983 with a full width at half maximum (FWHM, which is a factor of 1.67 larger than the Gaussian s parameter) of ~47 years, while the fit to the $NO_X$ emissions peaked in ~1995 with a FWHM of ~63 years. The shifts in the amplitude and the phase of the average simulated ozone seasonal cycle both reached peaks in

~1985, closely corresponding to the VOC emission peak. The FWHM of the ozone seasonal cycle amplitude shift (~48 years) also closely matches the FWHM of the VOC emissions. In contrast, the FWHM of the ozone seasonal cycle phase shift (~65 years) corresponds more closely to the FWHM of the $NO_X$ emissions. A simple hypothesis can provide a qualitative explanation for this correspondence. The VOC emissions provide fuel for photochemical production of ozone; thus these emissions exert primary control of the seasonal cycle amplitude driven by summertime production. The $NO_X$

emissions provide the catalyst that determines whether photochemistry produces or destroys ozone – once the $NO_X$ emissions are large enough that photochemical production dominates the seasonal cycle and moves the seasonal maximum into the summer, the phase shift ends, since the maximum cannot continue shifting into the autumn, and the seasonal maximum will not shift back until $NO_X$ emissions decrease to levels low enough that photochemical production no longer dominates the ozone budget. In summary, we are suggesting that the NOx emissions largely control the timing of seasonal

maximum in ozone, while the VOC emissions control the seasonal cycle amplitude. If this hypothesis is correct,



consideration of the role of biogenic VOCs could help to explain some of the diversity in the seasonal cycles and shifts seen among the model simulations; as can be seen in Figure 1 of Griffiths et al. (2021) the temporal variation of the biogenic VOCs emissions are significantly different across the models.

Assuming that the above hypotheses are correct, the ozone seasonal cycle shift derived from observations must reflect the time evolution of emissions, and thereby provide tests of the emission estimates upon which the model simulations are based. The measurement records (maximum of 44 years) are so short that the precision of the parameters of the seasonal cycle shift that can be derived from the measurements (see Table 3) is limited, as indicated by the relatively large confidence limits for those parameters included in Figure 8. However, two points can be noted. First, the average year of the maximum shift in the amplitude of the observed ozone seasonal cycle (1990 ± 3 years) is later than the  maximum of the VOC

emissions (1983 ± 1 year); since we expect these two maxima to be the same, this disagreement may indicate that anthropogenic VOC emissions actually peaked a few years later than indicated in the emission inventory. The uncertainty in the year of the maximum phase shift determined from observations (1985 ± 8 years) prevents a precise comparison between the emission maxima and the phase shift maxima. Second, the widths of the amplitude and phase shifts of the observed seasonal cycle (15 ± 9 and 17 ± 17 years, respectively) appear to be smaller than the widths of the $NO_X$ and VOC emissions

(38 ± 2 and 28 ± 2 years, respectively), but again the uncertainty of the observational determination prevents a firm conclusion.

The seasonal cycle of ozone reflects the annual variability of the sources and sinks of ozone; thus its accurate simulation is expected to present a stringent test for models. Given the paucity of the observational ozone record, both spatially but more importantly temporally, improved confidence in our understanding of changes in the seasonal ozone cycle must primarily

come from improved agreement between different model simulations. In this work we document relatively large seasonal cycle shifts that are common to the entire northern midlatitude baseline troposphere; given the magnitude of these shifts, which we attribute to changing precursor emissions throughout northern midlatitudes, it may be difficult to determine the impact of the changing climate (e.g., Fowler et al., 2008; Clifton et al., 2014) independently from the that of changing precursor emissions on the midlatitude ozone seasonal cycle.



**Data Availability:** All of the data utilized in this paper are available from public archives referenced in the paper.

**Author Contributions:** H.B. and D.D.P. designed the research and performed the analysis; S.E.B., K.T., M.D., N.O., F.M.O., L.H., T.W., and J.Z. performed model simulations; S.T.T. extracted model simulation results; H.B. and D.D.P. wrote the paper with input from all other authors.

**Competing interests:** The authors declare that they have no conflict of interest. Disclosure: David Parrish works as an atmospheric chemistry consultant (David D. Parrish, LLC); he has had contracts funded by several state and federal agencies and an industrial coalition, although they did not support the work reported in this paper.

**Acknowledgments**

Henry Bowman's efforts were supported by Carleton College's endowed internship funds, in particular from the Littell Endowed Internship Fund, designed to support undergraduate student internships with a focus in environmental studies. Steven Turnock would like to acknowledge that support for this work came from the UK-China Research and Innovation Partnership Fund through the Met Office Climate Science for Service Partnership (CSSP) China as part of the Newton Fund. Fiona M. O'Connor would like to acknowledge support from the BEIS and DEFRA Met Office Hadley Centre Climate Programme (GA01101). Makoto Deushi and Naga Oshima were supported by the Japan Society for the Promotion of Science KAKENHI (grant numbers: JP18H03363, JP18H05292, JP19K12312, and JP20K04070), the Environment Research and Technology Development Fund (JPMEERF20202003 and JPMEERF20205001) of the Environmental Restoration and Conservation Agency of Japan, the Arctic Challenge for Sustainability II (ArCS II), Program Grant Number JPMXD1420318865, and a grant for the Global Environmental Research Coordination System from the Ministry of the Environment, Japan (MLIT1753). Susanne Bauer and Kostas Tsigaridis acknowledge resources supporting this work were provided by the NASA High-End Computing (HEC) Program through the NASA Center for Climate Simulation (NCCS) at Goddard Space Flight Center. The CESM project is supported primarily by the National Science Foundation. Computing and data storage resources, including the Cheyenne supercomputer (doi:10.5065/D6RX99HX), were provided by the Computational and Information Systems Laboratory (CISL) at NCAR. NCAR is sponsored by the National Science Foundation.

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
