# Peer review of "Changes of Anthropogenic Precursor Emissions Drive Shifts of Ozone Seasonal Cycle throughout Northern Midlatitude Troposphere"

_Atmospheric Chemistry and Physics, 2021_

## Author Comment (AC1)

**Authors' Response to Comments of Anonymous Referees**

**Changes of Anthropogenic Precursor Emissions Drive Shifts of Ozone Seasonal Cycle throughout Northern Midlatitude Troposphere**

Henry Bowman, Steven Turnock, Susanne E. Bauer, Kostas Tsigaridis, Makoto Deushi, Naga Oshima, Fiona M. O'Connor, Larry Horowitz, Tongwen Wu, Jie Zhang, Dagmar Kubistin and David D. Parrish

We are grateful to the referees for the time and effort that they invested in reviewing our paper, and for their insightful comments, all of which are addressed below. This response has been prepared in consultation with all coauthors. In the following we reproduce the original referee comments in black regular font, and include our responses in *blue italic font*.

**Reply to Anonymous Referee #1 – RC1**

General comments:

The manuscript presents a comprehensive analysis of the shifts in timing and strength of the seasonal ozone cycles at remote locations in the Northern Hemisphere midlatitudes. The paper is within the scope of 'Atmospheric Chemistry and Physics' but – to my mind – requires some revisions prior to its acceptance for ACP. See my specific comments below.
*Thank you for your thorough review and insightful comments; some are particularly helpful for us in improving our original manuscript.*

Abstract, line 19: use different word from "amplitude" as it here represents the double of the amplitude as (mathematically) defined in equation 3 and shown in Fig. 5.
*Thank you. The sentence in question has been changed to "Beginning in ~1940 the magnitude of the seasonal cycle increased by ~10 ppb (measured from seasonal minimum to maximum), and the seasonal maximum shifted to later in the year by about 3 weeks."*

Line 103: Reider et al. needs to read Rieder et al., here and in the references.
*Thank you. That typo has been corrected throughout.*

Lines 146 – 154: as seen later in the results, the model output varies considerably among the different models. However, this paragraph provides hardly any information about the different models. There is reference made to the table in the supplementary material that provides some details but some short summary of the differences of the models might help to interpret differences in the model performance below. How about the spatial resolution of the models, number of vertical layers, different parametrization, different meteorological input etc.?
*Providing a cogent summary of the similarities and differences of these sophisticated and complex models is quite difficult, and outside the expertise of the lead authors of this paper. We believe that any attempt to interpret variation among the different model results will require an investigation that is much more detailed than can be included in a summary table. For example, Griffiths et al. (2021) have included such a summary as Table S1 of their Supplement, but full model details can only be obtained from the literature publications describing each model. We do give references to those publications (Table S4), but have not attempted to go beyond those references and the particularly relevant model difference that we included in our Table S2.*

Reference to the Earth System Grid Federation should be added in the acknowledgements.

*Thank you for this suggestion; the Earth System Grid Federation is now included in the acknowledgements.*

Are there no model simulations after 2014? Since the most dynamic part is the last 30-40 years, an extension to the most recent past would be appreciated.

*Thank you for this question. One element of the CMIP6 experiment (Eyring et al., 2016) included global climate model historical simulations from 1850 to the near present. Preparation for those simulations is time consuming, so the end of 2014 was chosen as the "near present"; thus, there are no comparable model simulations after 2014. The introduction to the CMIP6 experiment has been slightly expanded to clarify this issue.*

*It should be noted that the quantification of the ozone seasonal cycle shift depends most strongly on the period of rapid emission changes (c.f., Figure 7), so 1940 to 1980 is most important part of the record for our analysis. In contrast, over the last 30-40 years the emissions remained relatively constant.*

Table 1: I do not believe that O3 sondes are launched at Jungfraujoch. Please correct.

*We apologize for the lack of clarity in our discussion. Table 1 was intended to define the locations that are considered in the analysis between both model simulations and observational data. As the reviewer notes the sondes are not launched from Jungfraujoch; instead, the sonde data set is composed of measurement averages of sondes launched from Hohenpeissenberg, Uccle, and Payerne. That sonde data set is taken to be representative for the entire western Europe region, partially because it is averaged over these three locations and partially because of the zonal similarity in baseline ozone discussed by Parrish et al. (2020). As such, we compare sonde data collected from Hohenpeissenberg, Uccle, and Payerne with model simulation results from the free troposphere region at 5-6 km above Jungfraujoch. We were trying to communicate this information in Table 1.*

*To clarify the information, we have split Table 1 into two parts, the first describing locations considered in model simulations, and the second describing locations where observations were made.. Additionally, we have gone through the paper to clarify and correct some ambiguities in our descriptions. This improvement also addresses this reviewer's later comment regarding the Figure 4 caption.*

Caption, figure 2: add "simulated" that it reads "… time series of simulated monthly mean ozone concentrations …"

*Thank you for this suggestion; we have made this change.*

Caption, figure 3: write "Harmonic analysis of simulated preindustrial seasonal cycles …"

*Thank you; we have also made this suggested change.*

Line 366 ff.: the authors report "… quantitative differences in simulations among models …" but the authors do not provide any further ideas about the causes of the differences. An elaboration on this topic is highly requested.

*This request is very important, but unfortunately is far beyond what we can undertake in this paper through any rigorous analysis. The historical simulations, upon which our analysis is based, are only an initial step in the CMIP6 effort. A series of CMIP- Endorsed Model Intercomparison Projects (MIPs) are planned (Table 3 of Eyring et al., 2016 lists 21 MIPs) that may well elucidate causes of the differences noted in our analyses. We do not believe that it is appropriate for us to speculate further about the causes of the model differences in this paper.*

Figure 4 (legend) is somehow misleading as it makes the reader believing that observations above Jungfraujoch (FT 5-6 km) are available. Only the last sentence of the caption explains that the black curve is based on the mean of the European sonde measurements (which are made in Uccle, Hohenpeissenberg and Payerne (see lines 274-275)).

*Thank you. This has been clarified by giving more details in the caption; it is now consistent with the above response to the reviewer's earlier comment regarding Table 1.*

Why do you use the model output above the Alps (which might be more difficult to interpret due to the underlying topography and potentially associated issues in the models) when the three observational sites are largely located North of the Alps. Why not choose another grid box of the model that better represents the centre among the three sonde sites (which will be above a less complex terrain)? Again, what's the spatial resolution of the global models? How does the model topography (of the Alps) influence the model output?

*Thank you for these questions. The approach we chose to begin this work was to select a limited number of locations at which to carry out the inter-model and model-measurement comparisons. Measurements were available from three European alpine surface sites, and from three European sonde data sets. Our analysis of these measurement data (Parrish et al., 2020 and earlier work) convinced us that common long-term trends and seasonal cycles were seen at the same elevation across all of these data sets, so our choice of comparison location was not considered critical. We chose to compare all models above Jungfraujoch for model levels that included the elevation of the surface sites (~3.6 km), and at a higher elevation in the free troposphere (5 to 6 km) where observations show little shorter-term ozone variability due to minimal impact from both 1) lofting of polluted continental boundary layer air, and 2) from well-defined stratospheric intrusions. For comparison to the surface measurements, we originally looked at the model simulations from the lowest model level. However, due to the low spatial resolution of the underlying terrain in the simulations, the lowest model levels were often well below the measurement site elevation.*

*In answer to the posed questions, the spatial resolutions of the models vary between about 1°x1.25° and 2°x2.5°. We cannot definitively quantify how the model topography (of the Alps) influences the model simulations, but comparison of graphs in Figure 5 gives a good indication that there are not large influences – the simulations in **5a** and **5d** (from the higher FT altitudes) are similar to those in **5b** and **5e** (from lower surface site elevation). Thus, we do not believe that our choice of comparison location is an important confounding factor in our analysis.*

Line 454: "We average over all three European locations …"; write "… all three European sonde locations …" if this is meant here (i.e. Uccle, Hohenpeissenberg, Payerne).

*This sentence refers to results from the three European locations considered in model simulations: Hohenpeissenberg, Jungfraujoch at surface elevation, and the free troposphere between 5 and 6 km above Jungfraujoch. To clear up the ambiguity, we have added the phrase "model-simulated" so that it now reads "We average over all three model-simulated European locations."*

Figure 8: does it need panels c and d? To my understanding, the widths of the Gaussian fit are hard to interpret. The authors may consider removing c and d or adding some interpretation of the findings.

*Thank you for raising this question, but we believe that panels c and d of Figure 8 do convey important information, and deserve to be retained. The Gaussian function - $r*exp\{-((t-m)/s)^2$ - utilized in the fits has three parameters. Two, r and m, quantify the magnitude and time of the*

*maximum seasonal cycle shift, respectively. The width of the Gaussian, s, quantifies how long that shift persisted (i.e., it gives a measure of the starting and ending times relative to the time of the maximum). Thus, this parameter does provide an important point of comparison between the observed and simulated shifts and the time evolution of the precursor emissions that we conclude drive those shifts.*

Figure 8: there seems to be some kind of systematic pattern in the simulations results for the six models seen in panels a, c, and e (increase from left to right). Maybe this is by chance as the models are sorted alphabetically, but again, is it possible to explain some of the behavior of the different models? The width of the Gaussian fit seems to correlate with the maximum amplitude increase.

*Thank you for the insightful comment. Yes, we believe that the identified systematic pattern is a result of chance correlation between the alphabetical sorting. However, it is also apparent that there is a general correlation between the amplitudes and the widths of the Gaussian fits, such that models that simulate stronger shifts also simulate shifts with longer time spans. Although this is interesting to note, investigating the cause of this correlation in detail is beyond the scope of this paper. We do discuss aspects of this correlation in the 5th paragraph of the Discussion (i.e., paragraph beginning with the sentence: "On average, the parameters that quantify the shifts in the seasonal cycle correlate strongly with those that quantify the emissions.").*

Table 3, caption: write "... second two rows give parameters for fits to observed and model simulated seasonal ozone cycles" to make clear that the lower two lines refer to ozone while the upper two lines refer to ozone precursors. Why are the numbers for the simulations averaged over all six sites (which ones? I am confused) while the observations are averaged over Hohenpeissenberg and the European Alpine sites only?

*Thank you for bringing this to our attention. We have added the word "ozone" so that the second part of the second sentence reads: "...; second two rows give parameters for fits to observed and model simulated seasonal ozone cycles." This should clarify that the first two rows of the table refer to precursor emissions shifts and the second two rows of the table refer to ozone seasonal cycle shifts. We have also modified the lines between table rows, which were confusing.*

*To answer the questions posed: The numbers for the simulated seasonal cycle shift are averaged over the sites and altitudes mentioned in Table 1a: Hohenpeissenberg, Jungfraujoch, and the free troposphere between 5 and 6 km above Jungfraujoch in Europe; and the free troposphere between 0.9 and 1.2 km above Trinidad Head, Lassen Volcanic NP, and the free troposphere between 5 and 6 km above Trinidad Head in North America. We average over all these sites to obtain an overall mean picture of seasonal cycle shifts while minimizing the effects of local emissions or phenomena. Observational data are averaged only over Hohenpeissenberg and the European Alpine sites because those are the only observational data sets that display seasonal cycle shifts statistically significant at the 95% confidence interval.*

Discussions and conclusions are overall rather descriptive and do not provide many explanations. E.g. paragraph on lines 524 – 547, which starts "The model simulations exhibit large variability, both among models and locations …", doesn't hardly provide any conclusions for the reasons for the disagreements and the differences. Only the very last sentence of the paragraph briefly attempts to do so.

*Thank you for this comment. However, as noted above in response to the comment referring to Line 366 ff., we have no means to rigorously reach conclusions regarding the reasons for the disagreements and differences between model results. Very importantly, we do not believe that it*

*is appropriate for us to speculate in this regard in our paper. For example, we could point out that many aspects of the ozone formation is represented differently in the models; e.g. Section S6 of the Supplement highlights the challenges presented by natural ozone precursor emissions. Also, each model treats both overall quantity of VOCs and classification of VOCs differently. So, although there is a common total VOC emission input provided to all the models for biomass burning and anthropogenic emissions, these emissions are classified differently and mapped onto different chemical mechanisms in each model. Thus, there are significant differences in the chemistry taking place in each model in addition to differences in model dynamics, spatial resolution, etc. Derwent et al. (2021) discuss some impacts that arise in simulated tropospheric ozone between models based on only small differences in the treatment of light alkanes, so when many other VOC species are included, larger differences in tropospheric ozone may be expected to arise between models. In summary, there is a long list of model differences that could be identified, but without substantial sensitivity analysis for all of the models, we cannot conclude if any of these differences actually cause any of the disagreements or differences that we identify. In summary, we believe that is preferable to not speculate regarding the causes in the absence of strong analysis support.*

Lines 553-557: is this statement really needed?
*Thank you for the comment. We agree that the first sentence describing how the symmetry of the Gaussian fit does not capture the levelling-off of recent emissions can be omitted; it has been removed.*

I am wondering if some of the speculation (like on lines 571 ff. "… we hypothesize that the changing ozone precursor emissions is the cause of the shifts in the seasonal ozone cycle throughout northern midlatitudes." could be confirmed by looking into other world regions where the O3 precursors behave differently over time. The model output should be available globally, and there are some long-term ground-based O3 observations in pristine environments also available in tropical regions and the Southern Hemisphere. Also the SHADOZ program could provide a valuable dataset of O3 sonde observations for comparison with the model output.
*Thank you for this excellent suggestion; we should have thought to include such analysis in our submitted manuscript. We have responded by including Section S7 in the Supplement, which compares model simulations at the sites with the longest measurement records in the SH (Cape Grim, Australia) and the NH (Hohenpeissenberg). We have also included the principal investigator for the Hohenpeissenberg ozone data (Dagmar Kubistin) as a coauthor of the paper, since we now rely more heavily on that observational data set.*

*In the manuscript, the two sentences on lines 581-548 of our original submission have been changed to "The temporal correlation between the changes in emissions and the ozone seasonal cycle does not necessarily prove our hypothesis, but a comparison of ozone seasonal cycles between southern and northern mid-latitudes does support a causal relationship. As discussed and illustrated in Section S7 of the Supplement, we find no evidence of a significant shift in either the phase or magnitude of the ozone seasonal cycle at southern midlatitudes. The presence of a shift in the ozone seasonal cycle throughout northern midlatitudes and its absence at southern midlatitudes is as expected from our hypothesis, due to the much smaller anthropogenic ozone precursor emissions in the southern hemisphere. For reference, Figure 8 of Crippa et al. (2020) illustrates the dramatic difference in emissions from fossil fuel combustion between hemispheres." A sentence in the Abstract has also been revised to read: "We hypothesize that changing precursor emissions are responsible for the shift in the ozone seasonal cycle; this is*

*supported by the absence of such seasonal shifts in southern mid-latitudes where anthropogenic emissions are much smaller."*

Lines 606 ff. ("measurement records […] are so short that the precision […] of the seasonal cycle shift […] is limited"): the observation records used here stop in 2016 and 2017, respectively. Newer data are available in the public repositories. Why no newer data is used?
*It was difficult to decide exactly when to end the measurement record for comparison with the model results, which extend through 2014. We selected the same measurement periods that were considered in Parrish et al. (2020), since the results of that previous analysis were already published. There is an argument to made that the measurements considered should not extend past 2014, since that is the end of the model simulations to which the measurements are compared. However, from that previous measurement analysis effort we knew that this would not significantly change the analysis, but would decrease the analysis precision. As the reviewer notes, we could have extended the measurement record considered in order to improve the precision of the analysis, but this could introduce new, more recent influences into the analysis that the model simulations would not have been able to consider. Thus, we believe that our selection of observational period is appropriate.*

Lines 614-616: remove this statement as "… the uncertainty […] prevents a firm conclusion." .
*Thank you. This statement has been removed.*

Lines 622-624: remove statement on determining the impact of the changing climate? This wasn't discussed/mentioned at all in the manuscript.
*Thank you. We have modified this statement to make it more general and more specifically relevant to northern midlatitudes:*
*"…; given the magnitude of these shifts, which we attribute to changing precursor emissions, it may be difficult to independently determine the effects of other factors, e.g. changing climate (Fowler et al., 2008; Clifton et al., 2014), on the northern midlatitude ozone seasonal cycle."*
*We believe that this statement is now accurate and worthy of inclusion.*

Line 625: the data availability statement is insufficient. The statement refers to "public archives referenced in the paper" but neither table 1 nor chapter 2.4 provide any indication how the data were accessed and where interested reader may find the data.
*Thank you for identifying this omission. The data availability statement has been changed to:*
*"**Data Availability:** All of the data utilized in this paper are available from public archives; most are fully discussed and referenced in the Acknowledgments section of Parrish et al. (2020). One additional data set is included in this paper, the surface observations at Hohenpeissenberg, which are available from the Tropospheric Ozone Assessment Report data base (Schultz, M.G. et al. (2017)."*

**Reply to Anonymous Referee #2 – RC2**

Bowman et al. present a comprehensive analysis of changes in the ozone seasonal cycle across the northern hemisphere. The manuscript is generally well prepared and fits the scope of ACP. I suggest some revisions and adjustments to the presentation of the methods and results before consideration for publication in ACP by the Editor.
*Thank you for your thorough review and insightful comments; responding to them has helped us to improve our manuscript.*

Specific comments are given below:

**General Comments:**

Section 1: Several studies (e.g. Clifton et al., 2014; Schnell et al., 2016; Jaidan et al. 2018; Rieder et al., 2018) have reported on potential future changes in the ozone seasonal cycle due to changes in emissions/climate. Some reference to these findings should be made either in the introduction or in the conclusion section, especially given the concluding remarks in L620ff.
*Thank you for this helpful suggestion. We already included references to Clifton et al. (2014) and Rieder et al. (2018), but we now expand our discussion of these papers to include the hypothesized impacts of a warmer climate and increased methane concentrations on tropospheric ozone. In the same discussion, we also incorporate Schnell et al. (2016) and Jaidan et al. (2018), which were not previously referenced. This expanded discussion can be found in the Introduction section.*

Section 2.1: This section is important to the manuscript as it details the statistical treatment of the data sets. I think however, it could be presented in more concise form, e.g. keep the fundamental equations and narrative in the main manuscript and move other parts to the supplement.
*Thank you for these thoughts, but we suspect that this comment refers to Section 2.2 where we detail the statistical treatment of the data sets. We have reviewed the section and believe that all of the step-by-step description that we included was indeed necessary for the reader to follow both the analysis and the later discussion. We have gone through this section again, and made editorial changes for conciseness that reduced the length of the section by more than 20%.*

Section 2.4: Why do the authors limit the observational data to 2016/17? or if already shorter than available data is used do not align it to the same time frame of the historical ESM simulations?
*Thank you for this attentive question. Please see our response above to a similar comment made by Referee #1 with regard to Lines 606 ff.*

Section 3.2: provides a comparison of the preindustrial seasonal cycle as derived from different ESMs, the authors however do not provide any insights (or hypotheses) regarding the drivers of differences among ESMS and which estimates might be more reliable (e.g. overall amplitude differences of factor ~3, the differences in North American and European FT sites)
*This request is very important, but unfortunately is far beyond what we can undertake in this paper through any rigorous analysis, and we do not believe that it is appropriate to speculate.. For more details, please see our response above to similar comments made by Referee #1 with regard to Line 366 ff and to the Discussions and conclusions ... on lines 524 – 547.*

Section 3.3: I am not convinced that fitting Eq (5) to the comparatively short observational time series adds all that much besides maybe for the European Alpine and HP data sets.
*Thank you for the comment. It is true that only the European Alpine and Hohenpeissenberg observational data sets allow statistically significant Gaussian fits to the seasonal ozone cycle shifts, due to the relatively short time span of measurements at other locations. However, some of the other observational data sets do allow statistically significant linear fits to those shifts; this is because a linear fit requires determination of only 1 additional parameter (i.e., the slope) compared to the 3 parameters required for the Gaussian fit. We deem it useful to include those linear fits in order 1) to show the average amplitude and phase of the seasonal cycle over the measurement records, and 2) show that there are, in some data sets, significant shifts, even if*

*they cannot be quantified by the Gaussian fits; these results do provide useful comparisons for the model simulations.*

Section 3.4: Figure 8 and the text surrounding need some clarifications/modifications. It is not entirely clear what we learn from the width of the Gaussian in panels (c) and (d).
*Thank you for raising this point. We believe that panels c and d of Figure 8 convey important information and deserve to be retained. The Gaussian function - $r*exp\{-((t-m)/s)^2$ - utilized in the fits has three parameters. Two, r and m, quantify the magnitude and time of the maximum seasonal cycle shift, respectively. The width of the Gaussian, s, quantifies how long that shift persisted (i.e., it gives a measure of the starting and ending times relative to the time of the maximum). Thus, this parameter does provide an important point of comparison between the observed and simulated shifts and the time evolution of the precursor emissions that we conclude drive those shifts.*

Discussion and Conclusions:

i) I would like the authors to address the spread among models a bit more in this section. While overall the models agree on the timing of the shifts in the amplitude and phase of the seasonal cycle the authors highlight substantial variability across models and sampled locations which remains widely unaddressed.
*Thank you for this comment. We agree that this is a significant area for future work to address, but as is the case for the referee's comments about section 3.2 and the earlier comments of Referee #1, it is beyond the expertise of this paper's lead authors to fully address the differences in seasonal cycle shifts between model simulations without speculation.*

ii) The authors hypothesize that the main driver of changes in the seasonal cycle are changes in anthropogenic precursor emissions. Testing this hypothesis for other regions, with different emission patterns, in the NH (e.g. Asia) or SH would make an interesting addition to the manuscript.
*Thank you for this excellent suggestion; similar to our response above to a similar comment made by Referee #1 with regard to lines 571 ff. we should have thought to include such analysis in our submitted manuscript. We have responded by including Section S7 in the Supplement. Please see the earlier response for more details.*

**Minor Comments:**

Table 1: Jungfraujoch should be specified as surface site, as far as I am aware ozone sondes are in Switzerland only launched in Payerne.
*We apologize for the lack of clarity in our discussion. As we responded above to a similar comment made by Referee #1, we have clarified the information regarding locations of simulations and observations.*

Table 1: provide Trinidad Head and 200km West of Trinidad Head as last entries to separate observations from the additional model grid cell analyzed.
*Thank you for the suggestion. This should be addressed by our edits to Table 1 detailed above.*

L104: typo Reider àRieder
*Thank you. That typo has been corrected throughout.*

L 273: typo Zugspitse à Zugspitze

*Thank you. That typo has also been corrected.*

L 278: I assume this refers to Parrish et al. (2014, 2020), but it is not entirely clear which "studies of western European baseline ozone concentrations" are refereed to.
*Yes, this refers to Parrish et al. (2014, 2020). References been added to this sentence to clarify.*

Table 2: does not add to much in the main body of the manuscript, add to supplemental material
*Thank you for this comment. However, we think that it is important to compare the results of the present analysis with the results of previously published studies directly within the manuscript.*

Figures: throughout: when showing model data add "simulated"
*Thank you for this suggestion. We have added "simulated" when describing model output in figure captions where applicable.*

Figures: not all homogeneously labeled, compare e.g. order Fig. 6 and Fig. 8
*We apologize, but we don't understand this comment.*

Fig. 2: legend should show "red" not "yellow" color.
*Thank you; we have improved the legend to clearly show a "red" color.*

Figs. 5 and 6: I suggest moving the legends indicating model color coding outside the plotting frame for readability, do so also in the supplement
*Thank you for the suggestion, however, moving the legends would require us to decrease the size of the figures, which would hurt their readability. We have positioned the legends so that they do not obscure important parts of the figure.*

Figure 8: align sub panels, panels (f) and (e) are shifted compared to (a)-(d).
*Thank you for the suggestion. However, the panels (e) and (f) have different vertical axes, and the extra width of these panels is necessary to show the different axis labels.*

Figure 9: maybe color code model results based on colors chosen in Figs. 4,5,6
*Thank you for this comment. We presume this comment refers to Figure 8, but we have not changed the color code. The models are identified through annotations; in the discussion of this figure we already refer to colors of specific symbols, and adding other colors would be confusing.*

**Reference not cited in manuscript**

Derwent, R. G., Parrish, D. D., Archibald, A. T., Deushi, M., Bauer, S. E., Tsigaridis, K., Shindell, D., Horowitz, L. W., Khan M. A. H., and Shallcross, D. E.: Intercomparison of the Representations of the Atmospheric Chemistry of Pre-Industrial Methane and Ozone in Earth System and Other Global Chemistry-Transport Models, Atmos. Env., 248, https://doi.org/10.1016/j.atmosenv.2021.118248, 2021.

---

## Author Response (AR2)

**Authors' Response to Editor's Report**

**Changes of Anthropogenic Precursor Emissions Drive Shifts of Ozone Seasonal Cycle throughout Northern Midlatitude Troposphere**

Henry Bowman, Steven Turnock, Susanne E. Bauer, Kostas Tsigaridis, Makoto Deushi, Naga Oshima, Fiona M. O'Connor, Larry Horowitz, Tongwen Wu, Jie Zhang, Dagmar Kubistin and David D. Parrish

We are grateful to the editor and to the referees for the time and effort that they invested in their second reviews of our paper. This response has been prepared in consultation with all coauthors.

**Editor's Report**

Comments to the author:

Dear authors,

your revised submission have been re-evaluated by the reviewers. While many points have been adequately addressed, in particular referee 1 suggested that the issue of discussing possible reasons for discrepancies of model results is still not sufficiently addressed, and pointed to several co-authors on the author list should have this expertise. Referee 2 in a comment to the editor referred to a 'missed opportunity'. I therefore suggest that you integrate a paragraph on reasons for mismatch of model and observations trends into this paper. While I do not expect original new research, a reflection on model evaluations as found in the literature, and what it means for the subject of this paper would enrich the paper. I am looking forward to receive your revised manuscript.

**Author's response**

We are grateful for the suggestion. The last paragraph of our paper has been modified as indicated below, and an additional reference has been added. New discussion integrated into the new paragraph is indicated in red text:

The seasonal cycle of ozone reflects the annual variability of the sources and sinks of ozone; thus its accurate simulation is expected to present a stringent test for models. Given the paucity of the observational ozone record, both spatially but more importantly temporally, improved confidence in our understanding of changes in the seasonal ozone cycle must primarily come from improved agreement between different model simulations. Our analysis has focused on changes in anthropogenic and biomass burning emissions, which were prescribed from the same source to the extent possible for all models; however, there were differences in implementing the prescribed emissions into the models, mainly from VOCs due to individual requirements of the chemistry scheme within each model. In addition, the representation of natural emissions (e.g. biogenic VOCs emitted from vegetation) differed between individual models, giving variation in the natural to anthropogenic emission ratios between models. Thus, remaining differences in emissions between models may cause some of the inter-model differences. More generally,

Griffiths et al. (2020) suggest that differences in the simulation of ozone from CMIP6 models could be due to inter-model variations in the treatment of chemical and physical processes including dynamic transport, stratosphere-troposphere exchange, photolysis, deposition, convection and boundary-layer schemes. There is a need to go beyond direct model-observation comparison studies; for example, multi-model perturbed parameter ensembles can be used to intercompare the sensitivity of models to different input parameters and/or parameterizations (Wild et al., 2020). Notably, in this work we document relatively large seasonal cycle shifts that are common to the entire northern midlatitude baseline troposphere; given the magnitude of these shifts, which we attribute to changing precursor emissions, it may be difficult to independently determine the effects of other factors, e.g. changing climate (Fowler et al., 2008; Clifton et al., 2014), on the northern midlatitude ozone seasonal cycle.

**Reference not originally cited in manuscript**

Wild, O., Voulgarakis, A., O'Connor, F., Lamarque, J.-F., Ryan, E. M., and Lee, L.: Global sensitivity analysis of chemistry–climate model budgets of tropospheric ozone and OH: exploring model diversity, Atmos. Chem. Phys., 20, 4047–4058, https://doi.org/10.5194/acp-20-4047-2020, 2020.